health and disease and epidemiology, ecology

SARS-CoV-2, COVID-19, influenza, virus–virus interaction, mathematical modelling

**Author for correspondence:**
Matthieu Domenech de Cellès
e-mail: domenech@mpiib-berlin.mpg.de

# The pitfalls of inferring virus–virus interactions from co-detection prevalence data: application to influenza and SARS-CoV-2

Matthieu Domenech de Cellès[1], Elizabeth Goult[1], Jean-Sebastien Casalegno[2,3] and Sarah C. Kramer[1]

[1]Max Planck Institute for Infection Biology, Infectious Disease Epidemiology group, Chariteplatz 1, Campus Charité Mitte, 10117 Berlin, Germany
[2]Laboratoire de Virologie des HCL, IAI, CNR des virus à transmission respiratoire (dont la grippe) Hôpital de la Croix-Rousse F-69317, Lyon cedex 04, France
[3]Virpath, Centre International de Recherche en Infectiologie (CIRI), Université de Lyon Inserm U1111, CNRS UMR 5308, ENS de Lyon, UCBL F-69372, Lyon cedex 08, France

MDdC, 0000-0002-9302-4858; EG, 0000-0002-3354-1730; J-SC, 0000-0003-3271-9856; SCK, 0000-0002-6177-2309

There is growing experimental evidence that many respiratory viruses—including influenza and SARS-CoV-2—can interact, such that their epidemiological dynamics may not be independent. To assess these interactions, standard statistical tests of independence suggest that the prevalence ratio—defined as the ratio of co-infection prevalence to the product of single-infection prevalences—should equal unity for non-interacting pathogens. As a result, earlier epidemiological studies aimed to estimate the prevalence ratio from co-detection prevalence data, under the assumption that deviations from unity implied interaction. To examine the validity of this assumption, we designed a simulation study that built on a broadly applicable epidemiological model of co-circulation of two emerging or seasonal respiratory viruses. By focusing on the pair influenza–SARS-CoV-2, we first demonstrate that the prevalence ratio systematically underestimates the strength of interaction, and can even misclassify antagonistic or synergistic interactions that persist after clearance of infection. In a global sensitivity analysis, we further identify properties of viral infection—such as a high reproduction number or a short infectious period—that blur the interaction inferred from the prevalence ratio. Altogether, our results suggest that ecological or epidemiological studies based on co-detection prevalence data provide a poor guide to assess interactions among respiratory viruses.

## 1. Introduction

The pandemic of coronavirus disease 2019 (COVID-19), caused by the novel severe acute respiratory syndrome coronavirus 2 (SARS-CoV-2), has emphasized the persistent threat posed by respiratory viruses. In addition to SARS-CoV-2, other major respiratory viruses like influenza and the respiratory syncytial virus (RSV) cause a substantial burden every year, estimated at 78 million cases of lower respiratory infections and 130 000 associated deaths worldwide in 2016 [1]. As evidenced by the current and past pandemics, the large host range of respiratory viruses—and the correspondingly high risk of spillover from animals into humans—also makes them prime candidates for emergence of currently unknown 'diseases X' [2]. Interaction—here broadly defined as the ability of one pathogen to affect infection or disease caused by another pathogen—is an intriguing yet understudied aspect of respiratory viruses' biology [3]. Although different nomenclatures have been proposed [4],

such interactions can be classified according to their sign, either positive (synonymously, synergistic or facilitatory) or negative (synonymously, antagonistic or competitive). According to experimental evidence, various biological mechanisms exist which make either sign *a priori* plausible [4]. Examples include, in the case of positive interactions, upregulation of viral target receptors [5] or cell fusion [6]; and, in the case of negative interactions, blocking of viral replication caused by the interferon response [7,8]. Intriguingly, different respiratory viruses may have opposing effects on COVID-19, e.g., rhinoviruses can inhibit SARS-CoV-2 infection via the interferon response [8], while influenza A viruses can facilitate it via upregulation of ACE2, the cognate receptor of SARS-CoV-2 in human cells [5,9]. SARS-CoV-2 interactions may have far-reaching implications for predicting not only the future course of the COVID-19 pandemic, but also the indirect effects of non-COVID-19 vaccines on COVID-19 [10]. Indeed, vaccines against a target pathogen may also indirectly affect non-target pathogens that interact with this target pathogen—an effect expected (by the law of signs) to reduce the non-target pathogen burden in case of positive interactions, and to increase it in case of negative interactions [11–13].

Because of their relevance to epidemiology and public health, a natural question is how best to identify and estimate interactions between respiratory viruses. Arguably, challenge studies in animals or humans provide the strongest form of evidence, because they can pinpoint the within-host mechanisms of interaction in a controlled experimental setting. However, such studies remain scarce and, more generally, it is not easy to predict their consequences at the scale of human populations [14]. Hence, epidemiological studies—ideally informed by experimental evidence to narrow the search range of interacting pathogens—remain indispensable to assess interactions, but it is unclear whether methods commonly used in such studies are well-suited to this task.

In particular, recent studies of SARS-CoV-2 interactions used a test-negative design [15] to compare the risk of SARS-CoV-2 infection among those infected with another respiratory virus (e.g. influenza) to that among those uninfected [16–18]. The underlying idea is conceptually simple: if two (or more) viruses do not interact and circulate independently, then the frequency of co-detection estimated from cross-sectional data should be approximately equal to the product of each virus's detection frequency—conversely, any significant deviation from equality should indicate interaction. However, earlier epidemiological and ecological modelling studies have cautioned against seemingly intuitive statistics of interaction [14,19,20]. In fact, to our knowledge the validity of this study design has not yet been systematically tested for emerging or seasonal respiratory viruses.

In this study, we aimed to determine whether epidemiological studies based on co-detection prevalence data enabled reliable estimation of interactions between respiratory viruses. To do so, we designed a simulation study that built on a general epidemiological model of co-circulation of two emerging or seasonal respiratory viruses. We show that cross-sectional estimates of co-infection prevalence—interpreted either alone or in combination with estimates of single-infection prevalences—provide a poor guide to assess interaction. Hence, we argue that earlier epidemiological studies based on this design should be interpreted with caution and that further longitudinal studies will be needed to elucidate the epidemiological interactions of SARS-CoV-2 with other respiratory viruses.

## 2. Methods

### (a) Transmission model of viral co-circulation

We developed a deterministic model of circulation of two respiratory viruses, assumed to interact during the infectious period (i.e. the period of transmissible viral infection, denoted by $I$) or during a transient period following clearance of infection (denoted by $T$). According to experimental evidence, such interactions can result from an antiviral state caused by non-specific innate immune responses (such as the interferon response), which develop early during infection and can persist for a short period after clearance of infection [7]. By contrast, we did not model long-term interactions (effected, for example, by adaptive cross-immunity), which are less likely for different species of respiratory viruses [7]. The model was similar to that originally proposed by Shrestha *et al.* [14], with the addition of a latent period (denoted by $E$) and of a realistic distribution for the infectious period, modelled as a Gamma distribution with shape parameter 2 [21]. The transmission dynamic of each virus was, therefore, represented by a susceptible exposed infected temporary recovered (SEITR) model [22], where $S$ represents susceptible individuals and $R$ recovered individuals. Following Shrestha *et al.*, we used a double index notation to indicate the infection status with respect to each virus, e.g. $X_{SE}$ represents the proportion of individuals susceptible to virus 1 and exposed to virus 2. As we primarily focused on respiratory viruses that cause epidemics lasting a few months, we made the reasonable assumption of a constant, closed population.

The model was defined by a set of $6 \times 6 = 36$ ordinary differential equations, represented schematically in figure 1. The force of infection of each virus $i \in \{1, 2\}$ was given by:

$$\lambda_1(t) = \frac{R_1}{1 - r_{0,1}} \gamma_1 p_1(t)$$
$$p_1(t) = \sum_{x \in \Xi} [X_{I_a x}(t) + X_{I_b x}(t)]$$
$$\lambda_2(t) = \frac{R_2}{1 - r_{0,2}} \gamma_2 p_2(t)$$
$$p_2(t) = \sum_{x \in \Xi} [X_{x I_a}(t) + X_{x I_b}(t)]$$

where $\Xi = \{S, E, I_a, I_b, T, R\}$ is the set of state variables, $R_i$ is the reproduction number of virus $i$, $r_{0,i}$ the initial fraction immune to virus $i$, $1/\gamma_i$ the average infectious period of virus $i$ and $p_i(t)$ the prevalence of infection with virus $i$. Importantly, the parameter $R_i$ is best interpreted here as the initial reproduction number in a partially immune population, as opposed to the basic reproduction number (given by $R_i/(1 - r_{0,i})$ for virus $i$) in a fully susceptible population [23]. We also defined the prevalence of individuals co-infected (purple compartments in figure 1):

$$p_{12}(t) = X_{I_a I_a}(t) + X_{I_a I_b}(t) + X_{I_b I_a}(t) + X_{I_b I_b}(t).$$

### (b) Statistic to infer interaction from co-detection prevalence data

Standard statistical tests of independence suggest that the following prevalence ratio (PR):

$$\mathrm{PR}(t) = \frac{p_{12}(t)}{p_1(t) \times p_2(t)},$$

could be used to infer interaction [16–18]. Intuitively, a prevalence ratio above unity indicates that the frequency of co-detection is higher than that expected by chance, suggesting that co-infection is facilitated—that is, that the interaction is positive, i.e. synergistic [19,20]. Correspondingly, a prevalence ratio below unity would suggest a negative, or antagonistic interaction. In numerical applications, we calculated the prevalence ratio at the time of peak co-infection prevalence, $t_{\max} = \mathrm{argmax}_t p_{12}(t)$ (cf. electronic

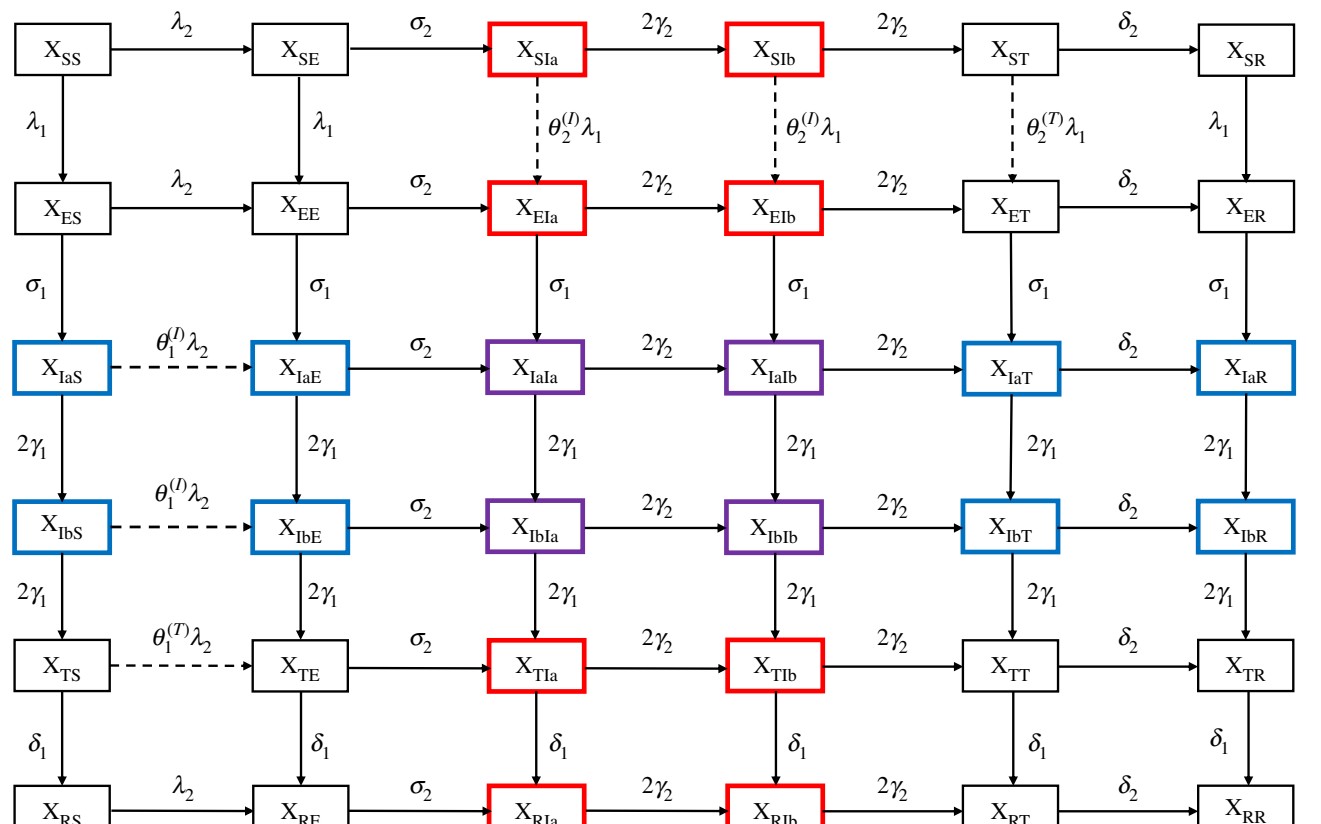

**Figure 1.** Schematic of epidemiological model of viral co-circulation. Individuals infectious with virus 1 are highlighted in blue, with virus 2 in red, and with both viruses in purple. Dashed lines indicate epidemiological transitions affected by interactions. (Online version in colour.)

supplementary material, figure S1), as we reasoned that empirical studies would have maximal statistical power to detect co-infection at that time point. Nevertheless, this choice is arbitrary, and we considered an alternative calculation in a sensitivity analysis, described below. In the following, we drop the time argument ($PR = PR(t_{max})$) and we simply refer to the prevalence ratio calculated at that time point.

Of note, other definitions of the prevalence ratio are possible and have been used in previous studies. For example, earlier studies of the association between SARS-CoV-2 and influenza compared the fraction of individuals infected with virus 2 among those infected with virus 1 to the fraction infected with virus 2 among those uninfected with virus 1—that is, a test-negative design [16–18]. Using the above notations and after some algebra, the corresponding prevalence ratio PR′ equals:

$$PR' = \frac{p_{12}/p_1}{(p_2 - p_{12})/(1 - p_1)} = PR \frac{1 - p_1}{1 - PR \times p_1}.$$

However, this alternative prevalence ratio is no longer symmetric in viruses 1 and 2, which implies an arbitrary choice of virus 1. We, therefore, prefer our formulation, but we point out that the two prevalence ratios are approximately equal for low prevalence of infection with virus 1. Furthermore, it can be shown that $PR' \geq 1 \Leftrightarrow PR \geq 1$, such that the sign of the interaction inferred from either ratio is identical.

### (c) Model parametrization

In numerical applications, we considered the pair influenza (virus 1)–SARS-CoV-2 (virus 2) and we fixed the parameters accordingly (table 1). Specifically, for influenza we assumed an average latent period of 1 day and an average infectious period of 4 days, resulting in an average generation time of 3 days [24,26]. For SARS-CoV-2, we assumed an average latent period of 4 days and an average infectious period of 5 days

(average generation time of 6.5 days) [25,27]. The reproduction number of influenza was fixed to 1.3 [23] and that of SARS-CoV-2 to 2.5 [25]. To initialize the model, we assumed that a small fraction $X_{ES}(0) = e_{0,1} = 10^{-3}$ had been exposed to influenza and $X_{SE}(0) = e_{0,2} = 10^{-5}$ to SARS-CoV-2. These initial conditions were chosen to reflect the epidemiological situation in early 2020 in Europe, where influenza was already circulating before the emergence of SARS-CoV-2 [28]. We further assumed that $X_{RS}(0) = r_{0,1} = 40\%$ of individuals were initially immune to influenza [29], and $X_{SR}(0) = r_{0,2} = 0\%$ to SARS-CoV-2. Other individuals were assumed fully susceptible ($X_{SS}(0) = 1 - e_{0,1} - e_{0,2} - r_{0,1} - r_{0,2}$), and all other compartments were initialized to 0. For simplicity, in the base models we considered only symmetric interactions, that is, the effect of virus 1 on virus 2 was assumed equal to that of virus 2 on virus 1. Furthermore, we assumed that interaction could not change sign over the course of infection, and we therefore tested negative ($0.2 \leq \theta^{(T)}, \theta^{(I)} \leq 1$) and positive ($1 \leq \theta^{(T)}, \theta^{(I)} \leq 5$) interactions separately.

### (d) Simulation protocol

In all scenarios, the model was integrated numerically for a period of 400 days, with state variable values recorded every 0.05 days.

### (e) Sensitivity analyses for influenza and SARS-CoV-2

To verify the robustness of our results, we conducted three sensitivity analyses. First, we considered an alternative prevalence ratio, similarly defined but averaged ±14 days around the time of peak co-infection prevalence. Second, although earlier experimental studies found that influenza can affect SARS-CoV-2 infection [5,9], the effect of SARS-CoV-2 on influenza infection, if any, is currently unknown. Previous experimental studies— e.g. of influenza and RSV [7]—demonstrated the possibility of non-symmetric interactions, where one virus affects the

**Table 1.** List of model parameters.

| parameter | meaning | fixed value or interval (influenza–SARS-CoV-2 analysis) | fixed value or interval (global sensitivity analysis) | source/comment |
|---|---|---|---|---|
| $\sigma_1^{-1}$ | average latent period of influenza | 1 day | $\sigma_1^{-1} = \sigma_2^{-1} = \sigma^{-1}$ $\sigma^{-1} \in [1-14]$ days | [24] |
| $\sigma_2^{-1}$ | average latent period of SARS-CoV-2 | 4 days | | [25] |
| $\gamma_1^{-1}$ | average infectious period of influenza | 4 days | $\gamma_1^{-1} = \gamma_2^{-1} = \gamma^{-1}$ $\gamma^{-1} \in [4-14]$ days | [26] |
| $\gamma_2^{-1}$ | average infectious period of SARS-CoV-2 | 5 days | | [25,27] |
| $R_1$ | reproduction number of influenza | 1.3 | $R_1 = R_2 = R$ $R \in \{1.5, 2.0, 2.5\}$ | [23] |
| $R_2$ | reproduction number of SARS-CoV-2 | 2.5 | | [25] |
| $e_{0,1}$ | initial fraction exposed to influenza | $10^{-3}$ | $e_{0,1} = e_{0,2} = 10^{-5}$ | assumption: influenza circulated before SARS-CoV-2 [28] |
| $e_{0,2}$ | initial fraction exposed to SARS-CoV-2 | $10^{-5}$ | | |
| $r_{0,1}$ | initial fraction immune to influenza | 0.4 | $r_{0,1} = r_{0,2} = r_0$ $r_0 \in [0.0-0.4]$ | [29] |
| $r_{0,2}$ | initial fraction immune to SARS-CoV-2 | 0 | | assumption |
| $\delta^{-1} = \delta_1^{-1} = \delta_2^{-1}$ | average post-infectious period | 1–14 days | 1–14 days | [30] |
| $\theta^{(I)} = \theta_1^{(I)} = \theta_2^{(I)}$ | strength of interaction during infectious period | 0.2–5 | 1–5 | assumption |
| $\theta^{(T)} = \theta_1^{(T)} = \theta_2^{(T)}$ | strength of interaction during post-infectious period | 0.2–5 | 1–5 | assumption |

other, but not the other way around. We therefore tested an alternative hypothesis of non-symmetric interactions, for which influenza affected SARS-CoV-2 infection, while SARS-CoV-2 did not affect influenza infection ($\theta_2^{(I)} = \theta_2^{(T)} = 1$). Third, we tested a scenario with seasonal transmission (representing, for example, weather-induced changes in virus survival [31,32]), modelled as a multiplicative effect on the transmission rate of each virus. For simplicity and interpretability, the seasonal forcing function was chosen as a sine wave:

$$s(t) = 1 + A\cos\omega(t - \phi),$$

where $\omega = 2\pi/365 \, \text{day}^{-1}$ is the annual angular frequency, $A \in \{0, 0.1, 0.2\}$ the peak relative semi-amplitude and $\phi = 100$ days the peak time. This peak time was chosen to approximately coincide with the peak time of co-infection (electronic supplementary material, figure S1), under the assumption of co-circulation during periods of higher seasonal transmission (e.g. during winter in temperate climates).

## (f) Global sensitivity analyses

To examine more generally the properties of viral infection and interaction that affected the prevalence ratio, we conducted a global sensitivity analysis for a broad range of respiratory viruses [33]. For simplicity, we assumed a fully symmetric model with identical characteristics of the two viruses, and we then proceeded in three steps. First, we used a Latin hypercube design to sample $10^3$ values (over the ranges indicated in table 1, [30]) of the following six parameters: average latent period ($1/\sigma$), average infectious period ($1/\gamma$), average post-infectious period ($1/\delta$), degree of interaction during the infectious period ($\theta^{(I)}$), degree of interaction during the post-infectious period ($\theta^{(T)}$) and initial fraction immune ($r_0$). Second, we simulated the model and calculated the prevalence ratio for every parameter set. Finally, we used a normal generalized additive regression model (GAM) with log-link to simultaneously estimate the association between the prevalence ratio and every input parameter [34]. For every parameter, the association was modelled using a basis of cubic splines, with a maximum basis

dimension of 10. Preliminary analyses indicated that the prevalence ratio was sensitive to the reproduction number, in isolation and in interaction with other parameters. To simplify the regression model, we, therefore, ran the global sensitivity analysis for three different values of the reproduction number (1.5, 2.0 and 2.5). To dissect the association of the prevalence ratio with every input parameter, we fitted the same GAM to the prevalence of co-infection ($p_{12}$, numerator of prevalence ratio) and to the product of single-infection prevalences ($p_1 p_2$, denominator of prevalence ratio).

## (g) Numerical implementation

We implemented and simulated all the models using the pomp package [35] in R v. 3.6.3 [36]. For the global sensitivity analysis, we used the mgcv package [34] to fit the GAMs and the ggeffects package [37] to plot the marginal effect of each input parameter. Finally, we used the renv package to keep track of all packages' version and to increase reproducibility [38].

# 3. Results

## (a) The prevalence ratio correctly identifies the sign, but not the degree, of uniform interactions

We first considered interactions of equal strength during the infectious and post-infectious periods ($\theta = \theta^{(I)} = \theta^{(T)}$)—henceforth referred to as *uniform* interactions. Example simulations of negative, neutral and positive interactions between influenza and SARS-CoV-2 are plotted in the electronic supplementary material, figure S1. Compared with the no-interaction scenario (peak co-infection prevalence: 0.2%), the peak amplitude of co-infection was lower for negative interaction (0.1%) and higher for positive interaction (1.3%). In all scenarios, however, the time of peak co-infection was approximately identical (range: 77.7–79.7 days). Next, we examined the general relationship between the strength of interaction and the prevalence ratio for different values of the post-infectious period in the range 1–14 days (figure 2). We found that the prevalence ratio equalled 1 for non-interacting viruses and thus permitted correct identification of neutral interactions ($\theta = 1$). For interacting viruses ($\theta \neq 1$), the prevalence ratio also correctly estimated the sign of the interaction, but systematically underestimated its strength. In addition, the degree of underestimation increased with the strength of interaction and the duration of the post-infectious period. Hence, we found evidence that the prevalence ratio enabled qualitative, but not quantitative, estimation of uniform interactions.

## (b) Higher interaction post-infection can cause the prevalence ratio to mis-identify non-uniform interactions

Next, we considered the more general case of interactions that differed during the infectious and the post-infectious periods, or *non-uniform* interactions ($\theta^{(I)} \neq \theta^{(T)}$). For these experiments, we assumed an average post-infectious period of 7 days and we tested negative ($0.2 \leq \theta^{(I)}, \theta^{(T)} \leq 1$) and positive ($1 \leq \theta^{(I)}, \theta^{(T)} \leq 5$) interactions separately. Because higher values of $\theta$ actually resulted in lower interaction when the true interaction was assumed negative, in the following we define the strength of interaction as $1 - \theta$ for negative interactions and as $\theta$ for positive interactions during either the infectious

or the post-infectious period. As shown in figure 3, we found that the prevalence ratio was a monotonic function of the strength of interaction during the infectious period, either decreasing for negative interactions or increasing for positive interactions. Hence, in either case stronger interaction during the infectious period helped the prevalence ratio identify the true interaction. By contrast, higher interaction during the post-infectious period blurred the interaction inferred from the prevalence ratio. For weak interaction during infection ($0.90 \leq \theta^{(I)} \leq 1.75$, 11% of tested scenarios for negative interactions and 14% for positive interactions), these two opposing effects combined caused the prevalence ratio to mis-identify the sign of interaction in scenarios with strong interaction post-infection. In the other scenarios, the prevalence ratio correctly identified the sign of the interaction, but substantially underestimated its strength (e.g. prevalence ratio of 0.56 for $\theta^{(I)} = 0.2$ and $\theta^{(T)} = 1$, of 1.91 for $\theta^{(I)} = 5$ and $\theta^{(T)} = 1$). These experiments demonstrate that the prevalence ratio is an unreliable measure of interaction between influenza and SARS-CoV-2.

## (c) Sensitivity analyses demonstrate the results' robustness for influenza and SARS-CoV-2

In sensitivity analyses, we first verified that our results were robust to an alternative calculation of the prevalence ratio (electronic supplementary material, figure S2). Second, we repeated our analyses for non-symmetric interactions with no effect of SARS-CoV-2 on influenza infection (electronic supplementary material, figure S3). The results were broadly comparable to those for symmetric interactions (figure 3), except that fewer parameter combinations caused the prevalence ratio to mis-identify the sign of interaction (7% of all combinations tested, compared with 13% for symmetric interactions). However, the strength of positive interaction was also more severely underestimated in this scenario (prevalence ratio range: 0.83–1.51, compared with 0.56–1.91 for symmetric interactions). Third, we tested a scenario with seasonal forcing in transmission (electronic supplementary material, figure S4). We found that the prevalence ratio was moderately sensitive to the amplitude of seasonality, with interaction more severely underestimated for higher amplitude. These results remained robust to alternative seasonal peak time values that approximately coincided with the time of peak co-infection.

## (d) Global sensitivity analysis highlights properties of viral infection that obscure or facilitate estimation of interaction

In a global sensitivity analysis of positive interactions ($\theta^{(I)}$, $\theta^{(T)} \geq 1$), we assessed how different properties of viral infection and interaction affected the prevalence ratio. As shown in figure 4, the prevalence ratio decreased with the average latent period, the average post-infectious period, the strength of interaction post-infection, and the initial fraction immune. Hence, these four parameters blurred the interaction inferred from the prevalence ratio. Conversely, the average length of, and the strength of interaction during the infectious period increased with the prevalence ratio and therefore facilitated estimation of the interaction. Of note, higher values of the reproduction number dampened all these variations and decreased the prevalence ratio. To better understand these

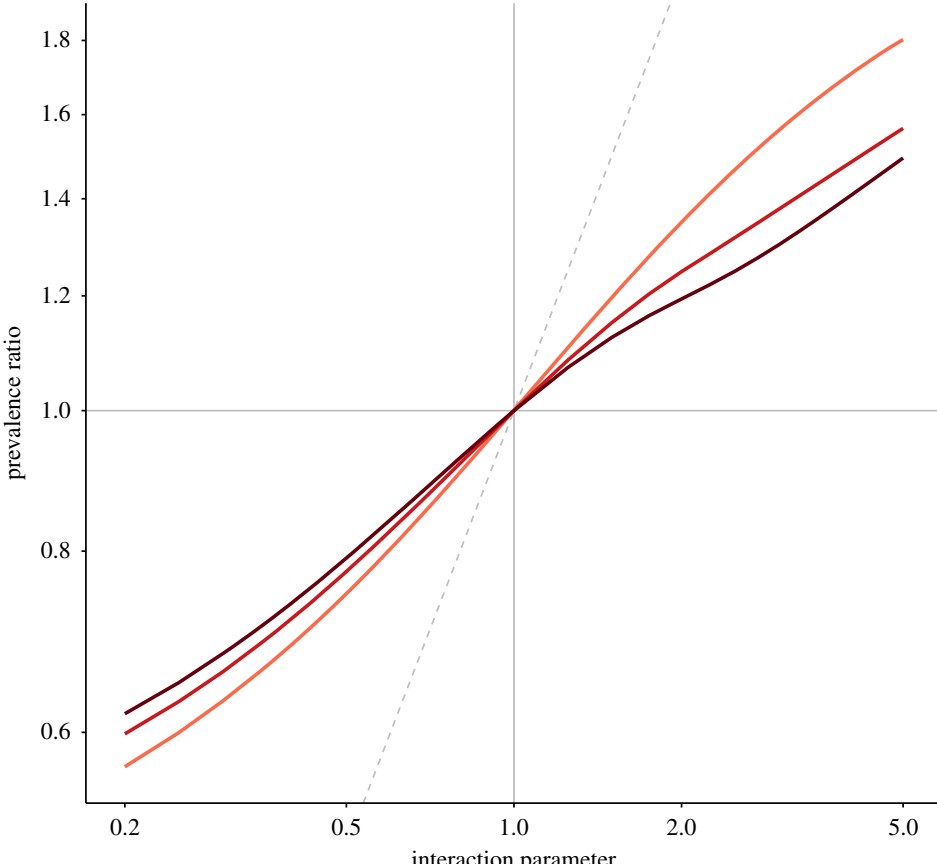

average post-infectious period (days) — 1 — 7 — 14

**Figure 2.** Relationship between strength of interaction and prevalence ratio for uniform interactions between influenza and SARS-CoV-2. The scenarios tested correspond to $\theta = \theta_1^{(I)} = \theta_1^{(T)} = \theta_2^{(T)} = \theta_2^{(I)}$ (x-axis), for three different values of the average post-infectious period ($1/\delta = 1/\delta_1 = 1/\delta_2$); other parameters were fixed to model the coupled dynamics of influenza and SARS-CoV-2 (cf. table 1). The dashed grey identity line depicts equality between the prevalence ratio and the true strength of interaction (PR $= \theta$). The x- and y-axes are log-transformed. (Online version in colour.)

associations, we estimated the effect of each parameter on the numerator (prevalence of co-infection $p_{12}$) and the denominator (product of single-infection prevalences $p_1 p_2$) of the prevalence ratio. As shown in the electronic supplementary material, figure S5, the prevalences of single- and co-infections decreased as the average latent period and the initial fraction immune increased. However, for both parameters co-infection prevalence decreased more rapidly than the product of single-infection prevalences, thereby explaining the overall variation of the prevalence ratio (figure 4). By contrast, the prevalences of single- and co-infections increased with all the other parameters—as expected, since these parameters either intensified ($\theta^{(I)}$ and $\theta^{(T)}$) or extended ($\gamma^{-1}$ and $\delta^{-1}$) (positive) interaction. The overall effect of these parameters on the prevalence ratio was again explained by their different effects on the prevalence of co-infection and on the product of single-infection prevalences. In summary, these results confirm our earlier experiments on influenza and SARS-CoV-2 and highlight additional factors that make it difficult to interpret the prevalence ratio as a measure of interactions between respiratory viruses.

## 4. Discussion

In this study, we aimed to determine whether the prevalence ratio—defined as the ratio of the prevalence of co-infection to the product of individual infection prevalences—enabled

reliable estimation of interactions between respiratory viruses. To do so we designed a simulation study that built on a broadly applicable epidemiological model of co-circulation of two emerging or seasonal respiratory viruses. By focusing on the pair influenza–SARS-CoV-2, we first demonstrated that the prevalence ratio systematically underestimated the strength of interaction, and could even mistake the sign of interactions that persisted after clearance of infection. In a global sensitivity analysis, we further identified properties of viral infection—such as a high reproduction number, a long latent period, or a short infectious period—that blurred the interaction inferred from the prevalence ratio. Our results show that, in the absence of precise information about the timing of interaction, ecological or epidemiological studies designed to estimate the prevalence ratio, or variations thereof, may be unreliable.

With the likely prospect of COVID-19 becoming endemic, there is a pressing need to elucidate the potential interactions of SARS-CoV-2 with other pathogens, in particular respiratory viruses. Thus far, most epidemiological studies of SARS-CoV-2 interaction used simple statistics of co-circulation, such as the prevalence of co-infection, the prevalence ratio, or some variation thereof [16–18,39,40]. As we showed here, however, such studies—even those carefully designed to control for various sources of bias like age or co-morbidities—are probably uninformative. Besides the prevalence ratio, we found that the prevalence of co-infection was also an unreliable measure of interaction, as low prevalences (less than or equal to 2.8% in the electronic supplementary material, figure S1, bottom

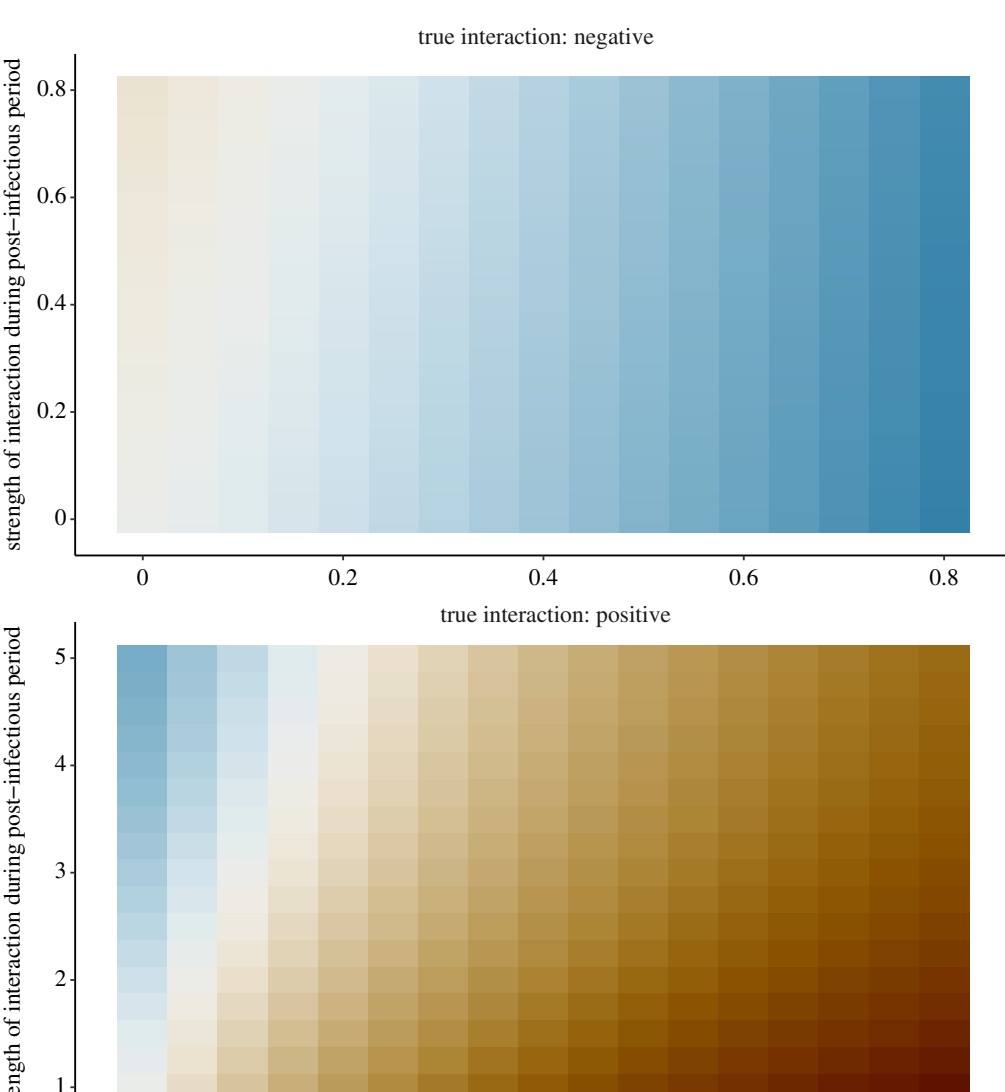

**Figure 3.** Relationship between strength of interaction and prevalence ratio for non-uniform interactions between influenza and SARS-CoV-2. The scenarios tested correspond to $\theta_1^{(I)} = \theta_2^{(I)} = \theta^{(I)}$ and $\theta_1^{(T)} = \theta_2^{(T)} = \theta^{(T)}$; other parameters were fixed to model the coupled dynamics of influenza and SARS-CoV-2 (cf. table 1). For negative interactions (top panel), the *x*-axis represents $1 - \theta^{(I)}$ and the *y*-axis $1 - \theta^{(T)}$; for positive interactions (bottom panel) $\theta^{(I)}$ and $\theta^{(T)}$. Hence, in either panel the true strength of interaction increases from left to right and from bottom to top. (Online version in colour.)

panel) could be consistent with strong, positive interactions. As suggested by our global sensitivity analysis, the deficiencies of statistics based on co-infection prevalence may be even more severe for SARS-CoV-2 infection, characterized by a relatively high reproduction number [25]. In summary, we submit that further epidemiological studies will be needed to elucidate the interactions of SARS-CoV-2 with other respiratory viruses.

More generally, our study adds to the growing body of evidence demonstrating the shortcomings of seemingly intuitive measures of interaction. Using the same model, Shrestha *et al.* demonstrated the unreliability of phase as an indicator of interaction [14]. Using a susceptible infected susceptible (SIS)-like model of multiple pathogens causing chronic infection, Hamelin *et al.* showed that the prevalence ratio (as defined in this study) exceeded unity for non-interacting pathogens [20]. By contrast, we found that the prevalence ratio equalled unity for non-interacting pathogens (figure 2). This discrepancy, explained by the different pathogens and

modelling assumptions considered in [20], highlights the sensitivity of the prevalence ratio to the characteristics of infection. More generally, it suggests that our results cannot be extrapolated to pathogens not well described by the susceptible infected recovered (SIR)-like model used here. Using a series of SIS and susceptible infected recovered susceptible (SIRS) models, Man *et al.* examined the properties of the odds ratio, defined as the ratio of the odds of one type in the presence of the other type, relative to the odds of this type in the absence of the other type—a quantity closely related to the alternative prevalence ratio PR′ defined above [19]. They proved that odds ratios exceeding unity could mask negative interactions. Despite differences in the scope of, and the models used in, this study, our results replicate this finding (figure 3). Furthermore, the association between the prevalence ratio and the interaction parameter in our study (figure 2) is comparable to that in [19] (fig. 2*a*; SIRS direct model). Finally, in a field study to assess

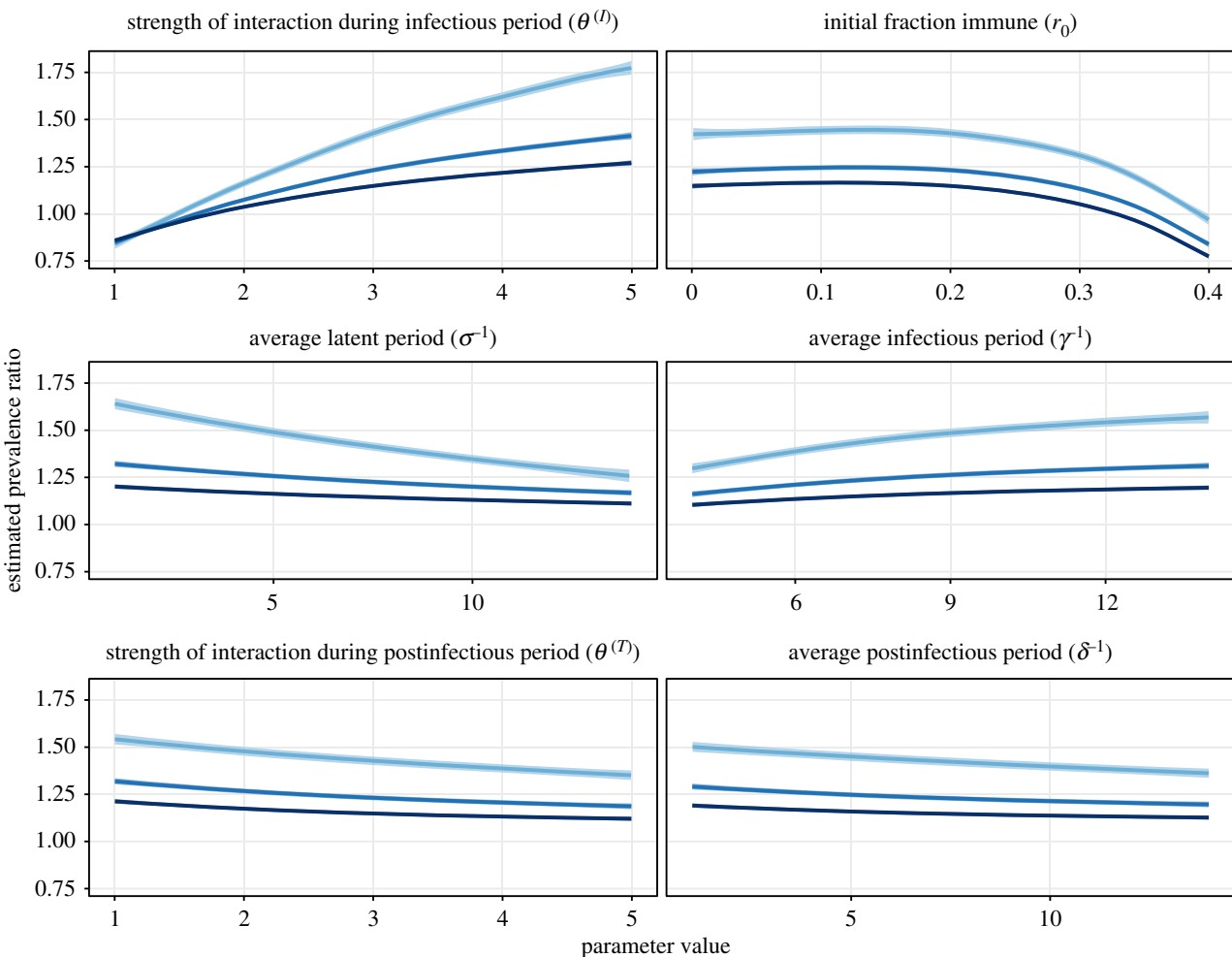

**Figure 4.** Global sensitivity analysis of prevalence ratio for positive virus–virus interactions. The association between the prevalence ratio and each input parameter was estimated using a GAM with cubic splines (sample size $n = 10^3$), for three different values of the initial reproduction number (1.5, 2.0 and 2.5). The corresponding adjusted $R$-squared equalled 97.4%, 97.5% and 97.9%. Each panel represents an input parameter, with the line (envelope) indicating the estimated (99% confidence interval) prevalence ratio while holding the other parameters fixed at $\sigma^{-1} = \gamma^{-1} = \delta^{-1} = 7$ days, $\theta^{(I)} = \theta^{(T)} = 3$ and $r_0 = 0.2$. For visual clarity, the $x$-axis values differ between panels (all periods are in days). (Online version in colour.)

interactions between an intestinal pathogen and nematodes in mice (where the true sign of interaction was known from previous experimental evidence), Fenton *et al.* reported that statistical methods based on cross-sectional data performed poorly and typically estimated the wrong sign of inter-action [41]. Our results align with these findings, and we second Fenton *et al.*'s caution against the use of such methods to study pathogen-pathogen interactions. In summary, our study broadly agrees with previous evidence, and provides new evidence specific to the epidemiology of emerging or seasonal respiratory viruses.

The shortcomings of the prevalence ratio demonstrated here might suggest the need for new statistical methods to estimate interaction from co-detection prevalence data. However, seconding Fenton *et al.* [41], we propose that methods based on longitudinal data—collected at an appropri-ately fine time scale—offer a more promising avenue of research. Among those methods, mathematical models of transmission provide a powerful tool to formulate and test bio-logically explicit mechanisms of interaction, while capturing the underlying, nonlinear dynamics of infection of each patho-gen [42]. Robust statistical inference techniques now facilitate fitting these models to epidemiological time series [43,44], as demonstrated by earlier successful applications in the

field of pathogen interactions [45–48]. Alternatively, advanced regression models have been developed to assess interactions between respiratory viruses [49], but such models may be lim-ited because they lack a mechanistic formulation of interaction. Altogether we propose that empirical or mechanistic models of longitudinal data will be required to study the interactions of SARS-CoV-2 with other respiratory viruses, and more gener-ally the interactions between respiratory pathogens [50].

Our study has four important limitations. First, because we used a deterministic model expressed in proportions, we side-stepped the important issue of statistical uncertainty, caused for example by finite sample size or imperfect measurement of infection prevalences. As the prevalence ratio was found to systematically underestimate the strength of interaction, such uncertainty—inevitable in practice—may further limit the ability of the prevalence ratio to correctly identify interactions. Second, for simplicity we did not include confounding variables (e.g. age) that may also affect estimation of the preva-lence ratio. Third, we considered only short-term interactions that rapidly waned after clearance of infection. Although such interactions appear to be the most biologically plausible for different species of respiratory viruses [7,8], long-term inter-actions resulting from adaptive cross-immunity have been documented and could be relevant to other systems, such as

the multiple types or subtypes of influenza [51,52]. Fourth, for simplicity we only modelled interactions that affected susceptibility to infection, because experimental evidence suggests this mechanism predominates among respiratory viruses [7,8]. However, other mechanisms—like changes in the transmissibility or the duration of infection—are biologically likely and could be tested for other classes of pathogens. Acknowledging all these limitations, our simple model could serve as a building block for further research on epidemiological interactions.

In conclusion, our results show that the inherently complex, nonlinear dynamic of emerging and seasonal respiratory viruses makes the interpretation of seemingly intuitive measures of interaction difficult, if not impossible. Despite these pitfalls, other statistical or mathematical methods based on longitudinal data should enable epidemiological research on pathogen interactions. Indeed, with increasing evidence that SARS-CoV-2 and other pathogens do not circulate in isolation but within polymicrobial systems, such research should remain a priority.

Data accessibility. All R programming codes are freely available from Edmond, the Open Data Repository of the Max Planck Society: https://doi.org/10.17617/3.8k.

Authors' contributions. M.D.d.C.: conceptualization, formal analysis, methodology, software, visualization, writing—original draft, writing—review and editing; E.G.: conceptualization, methodology, writing—review and editing; J.-S.C.: conceptualization, methodology, writing—review and editing; S.C.K.: conceptualization, methodology, writing—review and editing. All authors gave final approval for publication and agreed to be held accountable for the work performed therein.

Competing interests. M.D.d.C. received postdoctoral funding (2017–2019) from Pfizer and consulting fees from GSK. All other authors declare no competing interests.

Funding. Open access funding provided by the Max Planck Society.

Acknowledgements. We thank Laura Barrero and Michael Briga for helpful comments on the manuscript.

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
