## [Peer Review File · Proceedings of the Royal Society B: Biological Sciences]

Review History

RSPB-2021-1835.R0 (Original submission)

Review form: Reviewer 1

Recommendation

Major revision is needed (please make suggestions in comments)

Scientific importance: Is the manuscript an original and important contribution to its field?

Good

General interest: Is the paper of sufficient general interest?

Good

Quality of the paper: Is the overall quality of the paper suitable?

Good

Is the length of the paper justified?

Yes

Should the paper be seen by a specialist statistical reviewer?

No

Do you have any concerns about statistical analyses in this paper? If so, please specify them explicitly in your report.

No

It is a condition of publication that authors make their supporting data, code and materials available - either as supplementary material or hosted in an external repository. Please rate, if applicable, the supporting data on the following criteria.

Is it accessible?

Yes

Is it clear?

Yes

Is it adequate?

Yes

Do you have any ethical concerns with this paper?

No

Comments to the Author

The reviewed article focused on the ability for co-detection prevalence data to inform virus-virus interactions, using a pair of viruses that cause seasonal epidemics (influenza) and a current pandemic (SARS-CoV-2). Research has shown that respiratory viruses interact, leading to both positive and negative interactions, which can alter dynamics of co-infections. Early epidemiological studies investigating viral interactions between SARS-CoV-2 and a variety of respiratory viruses used the prevalence ratio to estimate the direction, and more importantly, strength of interactions among viruses. However, there is limited support for prevalence ratios to accurately estimate interactions among viruses (and all parasites). This study utilized simulations of a SEITR disease model with varying transmission parameters to test whether the test-negative design (co-detection prevalence data) is valid for testing for positive/negative interactions between respiratory viruses. The authors state that their results show that caution should be taken when interpreting epidemiological studies of SARS-CoV-2 (and other viruses) that use the test-negative design to infer directionality and magnitude of viral interactions.

The authors provide a clear background of viral interactions as well as current/past approaches to investigate interactions between SARS-CoV-2 and other respiratory viruses. The results of this study are not surprising, given the work of Fenton et al. 2014, but are exceedingly important to be reiterated, as early SARS-CoV-2 manuscripts make broad claims that may not be supported using the prevalence ratio, as shown here. I have one major concern regarding the interpretation of the relationship between the prevalence ratio and the authors' measure of interaction strength (θ , theta), but find the rest of the manuscript and study to be satisfactory. The manuscript is well organized and well written, and the figures are clear and easy to interpret.

Comments

Overall: The authors discuss the limitations of the prevalence ratio in terms of its ability to estimate strength of interactions. Specifically, the authors state that "because of a concave association, under-estimation became more severe as the strength of the interaction increased". While this concave relationship is apparent and causes the inherent underestimation of interaction strength, the concavity of this relationship is caused not by the prevalence ratio, but rather your measure of interaction strength (θ , theta), because θ (theta) itself is not symmetrical. Specifically, negative interactions are inherently bound between 0 and 1, while positive interactions are all numbers above 1. Therefore, a natural log transformation of θ results in a symmetrical distribution of interactions and would be akin to a log response ratio (sensu a meta-analysis). A natural log transformation of a 2x positive interaction (θ (theta) = 2) is 0.693, while a natural log transformation of a 2x negative interaction (θ (theta) = 0.5) is -0.693; and natural log

transformation of unity (1) is 0. A 1:1 slope (and intercept of 1) would then indicate whether the prevalence ratio accurately measures strength of interactions among viruses.

I understand the purpose of this paper is to highlight the shortcomings of the PR in inferring interactions, *sensu* Fenton et al 2014, and I generally agree with the authors' conclusions that PR is not as informative as it seems and, instead, longitudinal studies are needed. However, the authors need to address the ability of the PR to accurately predict sign and strength of interactions after remediating the non-symmetry of their measure of interaction strength (θ , theta).

Line 22-23: Is there evidence that SARS-CoV-2 causes seasonal epidemics? Currently, still in pandemic, with potential for seasonal epidemic, but not sure there is enough evidence to argue this at this point.

Lines 49-51: Please clarify the directionality of the suggested indirect effects of vaccines on non-target pathogens.

Lines 142-146: You should also test a second alternative hypothesis of non-symmetric, allow the direction of the effect of SARS-CoV-2 on influenza to be the opposite sign of the effect of influenza on SARS-CoV-2.

Line 137: Change 5×10^{-2} to 0.05 days.

Line 171-172: The authors need to provide mean \pm se for the "peak time" of co-infection across scenarios.

Line 203: Only the positive interaction is "severely under-estimated" in this scenario, the negative interaction may be over-estimated, or possibly accurately estimated. But see general comment above regarding assessing the PR ability to estimate interaction strength.

Line 212-216: The authors posit that parameters in their SEITR model may differentially affect single- and co-infections and state that parameters associated with interactions post-infection affect single-infection prevalences more rapidly and strongly than co-infection, but provide no evidence for these statements. The authors need to present the relationships among these post-infection parameters and single-infection prevalences in conjunction with those of co-infection prevalence to then compare effects of these parameters quantitatively on the single- and co-infection endpoints. Or, at least, move these lines to the discussion, as they seem more like assumptions than supported by evidence within this manuscript.

Line 237: define "low" prevalence

Line 238-240: What deficiencies do you refer to? Low prevalence ratios? Further, the sensitivity analysis indicates that for a virus with a higher reproduction number (e.g., SARS-CoV-2) there is no relationship between prevalence ratio and latent period, suggesting that latent period may not affect co-infection. Does this suggest that the length of latent period has no effect on co-infection prevalence?

Line 259: Your results indicate that PR does more often than not predict the direction of interaction. Please provide estimates of the percent of runs in which PR accurately predicted the direction of interactions (e.g., PR correctly identified the direction of 70% of positive interactions, but only 50% of negative interactions).

Figure 5: For ease of interpretation, provide symbols with titles on each panel. The text consistently refers to these parameters by their symbol, rather than what each parameter corresponds with. Even with Table 1, it takes effort to match symbols with parameter definitions. Additionally, these lines represent the relationship between PR and each input parameter, as estimated via a GAM. As such, these figures either need data points or error bars from the GAMs to provide a measure of model accuracy and clarity that these lines are from statistical models, not the SEITR model.

Review form: Reviewer 2

Recommendation

Major revision is needed (please make suggestions in comments)

Scientific importance: Is the manuscript an original and important contribution to its field?

Good

General interest: Is the paper of sufficient general interest?

Good

Quality of the paper: Is the overall quality of the paper suitable?

Good

Is the length of the paper justified?

Yes

Should the paper be seen by a specialist statistical reviewer?

No

Do you have any concerns about statistical analyses in this paper? If so, please specify them explicitly in your report.

No

It is a condition of publication that authors make their supporting data, code and materials available - either as supplementary material or hosted in an external repository. Please rate, if applicable, the supporting data on the following criteria.

Is it accessible?

N/A

Is it clear?

N/A

Is it adequate?

N/A

Do you have any ethical concerns with this paper?

No

Comments to the Author

This paper is focusing on the possibility to use co-detection ratio as a proxy for inferring pathogen/pathogen interactions, with a specific focus on Sars-Cov-2 and influenza. Briefly, the authors shown that this method is not really reliable. I think this is an interesting paper that would deserve to be published once my comments would have been addressed.

My main concern is about the context simulated, which is focusing actually on two emerging pathogens (since only transient dynamics is considered). However, even if influenza is more complicated than that, we cannot really assume that everybody is susceptible to influenza. More broadly, it rises the question about the validity of these results when considering endemic pathogens instead of emerging pathogens. This is very important to clarify regarding the strength of the messages ("In conclusion, our results show that the inherently complex, non-linear dynamic of respiratory viruses makes the interpretation of seemingly intuitive measures of interaction difficult, if not impossible.") and the pathogens studied as a case-study (sars-cov-2 and influenza).

My other concern is about the model itself, especially the lack of transmission seasonality (not required for emerging pathogens, but could be needed for endemic ones) and the fact that stochasticity is not considered (but since the co-detection method does not seem to work very well, adding stochasticity will probably makes this result even stronger). I think these two important features need to be, at least, discussed.

Decision letter (RSPB-2021-1835.R0)

05-Oct-2021

Dear Mr Domenech de Cellès:

I am writing to inform you that your manuscript RSPB-2021-1835 entitled "The pitfalls of inferring virus-virus interactions from co-detection prevalence data: Application to influenza and SARS-CoV-2" has, in its current form, been rejected for publication in Proceedings B.

This action has been taken on the advice of referees, who have recommended that substantial revisions are necessary. With this in mind we would be happy to consider a resubmission, provided the comments of the referees are fully addressed. However please note that this is not a provisional acceptance.

Sincerely,
Professor Hans Heesterbeek
<mailto:proceedingsb@royalsociety.org>

Associate Editor
Board Member: 1
Comments to Author:

Following two expert reviews, there is consensus that the manuscript submitted by Domenech de Cellès et al. provides important insights to the limitations of current methods to measuring virus-virus interactions. This is undoubtedly an important topic. All reviewers agree that the manuscript would benefit from major revision. In preparing a revised manuscript for resubmission, please address carefully each point raised by the reviewers.

Reviewer(s)' Comments to Author:

Referee: 1

Comments to the Author(s)

The reviewed article focused on the ability for co-detection prevalence data to inform virus-virus interactions, using a pair of viruses that cause seasonal epidemics (influenza) and a current pandemic (SARS-CoV-2). Research has shown that respiratory viruses interact, leading to both positive and negative interactions, which can alter dynamics of co-infections. Early epidemiological studies investigating viral interactions between SARS-CoV-2 and a variety of respiratory viruses used the prevalence ratio to estimate the direction, and more importantly, strength of interactions among viruses. However, there is limited support for prevalence ratios to accurately estimate interactions among viruses (and all parasites). This study utilized simulations of a SEITR disease model with varying transmission parameters to test whether the test-negative design (co-detection prevalence data) is valid for testing for positive/negative interactions between respiratory viruses. The authors state that their results show that caution should be taken when interpreting epidemiological studies of SARS-CoV-2 (and other viruses) that use the test-negative design to infer directionality and magnitude of viral interactions.

The authors provide a clear background of viral interactions as well as current/past approaches to investigate interactions between SARS-CoV-2 and other respiratory viruses. The results of this study are not surprising, given the work of Fenton et al. 2014, but are exceedingly important to be reiterated, as early SARS-CoV-2 manuscripts make broad claims that may not be supported using the prevalence ratio, as shown here. I have one major concern regarding the interpretation of the relationship between the prevalence ratio and the authors' measure of interaction strength (θ), but find the rest of the manuscript and study to be satisfactory. The manuscript is well organized and well written, and the figures are clear and easy to interpret.

Comments

Overall: The authors discuss the limitations of the prevalence ratio in terms of its ability to estimate strength of interactions. Specifically, the authors state that "because of a concave association, under-estimation became more severe as the strength of the interaction increased". While this concave relationship is apparent and causes the inherent underestimation of interaction strength, the concavity of this relationship is caused not by the prevalence ratio, but rather your measure of interaction strength (θ , theta), because θ (theta) itself is not symmetrical. Specifically, negative interactions are inherently bound between 0 and 1, while positive interactions are all numbers above 1. Therefore, a natural log transformation of θ results in a symmetrical distribution of interactions and would be akin to a log response ratio (sensu a meta-analysis). A natural log transformation of a 2x positive interaction (θ (theta) = 2) is 0.693, while a natural log transformation of a 2x negative interaction (θ (theta) = 0.5) is -0.693; and natural log transformation of unity (1) is 0. A 1:1 slope (and intercept of 1) would then indicate whether the prevalence ratio accurately measures strength of interactions among viruses.

I understand the purpose of this paper is to highlight the shortcomings of the PR in inferring interactions, sensu Fenton et al 2014, and I generally agree with the authors' conclusions that PR is not as informative as it seems and, instead, longitudinal studies are needed. However, the authors need to address the ability of the PR to accurately predict sign and strength of interactions after remediating the non-symmetry of their measure of interaction strength (θ , theta).

Line 22-23: Is there evidence that SARS-CoV-2 causes seasonal epidemics? Currently, still in pandemic, with potential for seasonal epidemic, but not sure there is enough evidence to argue this at this point.

Lines 49-51: Please clarifying the directionality of the suggested indirect effects of vaccines on non-target pathogens.

Lines 142-146: You should also test a second alternative hypothesis of non-symmetric, allow the direction of the effect of SARS-CoV-2 on influenza to be the opposite sign of the effect of influenza on SARS-CoV-2.

Line 137: Change 5×10^{-2} to 0.05 days.

Line 171-172: The authors need to provide mean \pm se for the "peak time" of co-infection across scenarios.

Line 203: Only the positive interaction is “severely under-estimated” in this scenario, the negative interaction may be over-estimated, or possibly accurately estimated. But see general comment above regarding assessing the PR ability to estimate interaction strength.

Line 212-216: The authors posit that parameters in their SEITR model may differentially affect single- and co-infections and state that parameters associated with interactions post-infection affect single-infection prevalences more rapidly and strongly than co-infection, but provide no evidence for these statements. The authors need to present the relationships among these post-infection parameters and single-infection prevalences in conjunction with those of co-infection prevalence to then compare effects of these parameters quantitatively on the single- and co-infection endpoints. Or, at least, move these lines to the discussion, as they seem more like assumptions than supported by evidence within this manuscript.

Line 237: define “low” prevalence

Line 238-240: What deficiencies do you refer to? Low prevalence ratios? Further, the sensitivity analysis indicates that for a virus with a higher reproduction number (e.g., SARS-CoV-2) there is no relationship between prevalence ratio and latent period, suggesting that latent period may not affect co-infection. Does this suggest that the length of latent period has no effect on co-infection prevalence?

Line 259: Your results indicate that PR does more often than not predict the direction of interaction. Please provide estimates of the percent of runs in which PR accurately predicted the direction of interactions (e.g., PR correctly identified the direction of 70% of positive interactions, but only 50% of negative interactions).

Figure 5: For ease of interpretation, provide symbols with titles on each panel. The text consistently refers to these parameters by their symbol, rather than what each parameter corresponds with. Even with Table 1, it takes effort to match symbols with parameter definitions. Additionally, these lines represent the relationship between PR and each input parameter, as estimated via a GAM. As such, these figures either need data points or error bars from the GAMs to provide a measure of model accuracy and clarity that these lines are from statistical models, not the SEITR model.

Referee: 2

Comments to the Author(s)

This paper is focusing on the possibility to use co-detection ratio as a proxy for inferring pathogen/pathogen interactions, with a specific focus on Sars-Cov-2 and influenza. Briefly, the authors shown that this method is not really reliable. I think this is an interesting paper that would deserve to be published once my comments would have been addressed.

My main concern is about the context simulated, which is focusing actually on two emerging pathogens (since only transient dynamics is considered). However, even if influenza is more complicated than that, we cannot really assume that everybody is susceptible to influenza. More broadly, it rises the question about the validity of these results when considering endemic pathogens instead of emerging pathogens. This is very important to clarify regarding the strength of the messages ("In conclusion, our results show that the inherently complex, non-linear dynamic of respiratory viruses makes the interpretation of seemingly intuitive measures of interaction difficult, if not impossible.") and the pathogens studied as a case-study (sars-cov-2 and influenza).

My other concern is about the model itself, especially the lack of transmission seasonality (not required for emerging pathogens, but could be needed for endemic ones) and the fact that stochasticity is not considered (but since the co-detection method does not seem to work very well, adding stochasticity will probably makes this result even stronger). I think these two important features need to be, at least, discussed.

Author's Response to Decision Letter for (RSPB-2021-1835.R0)

See Appendix A.

RSPB-2021-2358.R0

Review form: Reviewer 1

Recommendation

Accept as is

Scientific importance: Is the manuscript an original and important contribution to its field?

Excellent

General interest: Is the paper of sufficient general interest?

Excellent

Quality of the paper: Is the overall quality of the paper suitable?

Excellent

Is the length of the paper justified?

Yes

Should the paper be seen by a specialist statistical reviewer?

No

Do you have any concerns about statistical analyses in this paper? If so, please specify them explicitly in your report.

No

It is a condition of publication that authors make their supporting data, code and materials available - either as supplementary material or hosted in an external repository. Please rate, if applicable, the supporting data on the following criteria.

Is it accessible?

Yes

Is it clear?

Yes

Is it adequate?

Yes

Do you have any ethical concerns with this paper?

No

Comments to the Author

The authors have adequately addressed the concerns of both reviewers. The changes made by the authors have improved the clarity of the writing and figures; the realism of their simulations by adding pre-existing immunity to influenza; and the impact of their results through the additional seasonal sensitivity analysis. The manuscript and context are important for both the scientific

community and for public health, as their results imply that studies using test-negative designs to infer directionality and magnitude of viral interactions may be flawed under certain conditions.

Review form: Reviewer 2

Recommendation

Accept as is

Scientific importance: Is the manuscript an original and important contribution to its field?

Excellent

General interest: Is the paper of sufficient general interest?

Excellent

Quality of the paper: Is the overall quality of the paper suitable?

Excellent

Is the length of the paper justified?

Yes

Should the paper be seen by a specialist statistical reviewer?

No

Do you have any concerns about statistical analyses in this paper? If so, please specify them explicitly in your report.

No

It is a condition of publication that authors make their supporting data, code and materials available - either as supplementary material or hosted in an external repository. Please rate, if applicable, the supporting data on the following criteria.

Is it accessible?

Yes

Is it clear?

Yes

Is it adequate?

Yes

Do you have any ethical concerns with this paper?

No

Comments to the Author

Sorry for the delay. I agree with the author's answers and I am therefore happy to recommend the publication of this manuscript.

Decision letter (RSPB-2021-2358.R0)

29-Nov-2021

Dear Mr Domenech de Cellès

I am pleased to inform you that your Review manuscript RSPB-2021-2358 entitled "The pitfalls of inferring virus-virus interactions from co-detection prevalence data: Application to influenza and SARS-CoV-2" has been accepted for publication in Proceedings B.

The referees do not recommend any further changes. Therefore, please proof-read your manuscript carefully and upload your final files for publication. Because the schedule for publication is very tight, it is a condition of publication that you submit the revised version of your manuscript within 7 days. If you do not think you will be able to meet this date please let me know immediately.

To upload your manuscript, log into <http://mc.manuscriptcentral.com/prsb> and enter your Author Centre, where you will find your manuscript title listed under "Manuscripts with Decisions." Under "Actions," click on "Create a Revision." Your manuscript number has been appended to denote a revision.

You will be unable to make your revisions on the originally submitted version of the manuscript. Instead, upload a new version through your Author Centre.

1) A text file of the manuscript (doc, txt, rtf or tex), including the references, tables (including captions) and figure captions. Please remove any tracked changes from the text before submission. PDF files are not an accepted format for the "Main Document".

2) A separate electronic file of each figure (tiff, EPS or print-quality PDF preferred). The format should be produced directly from original creation package, or original software format. Please note that PowerPoint files are not accepted.

3) Electronic supplementary material: this should be contained in a separate file from the main text and the file name should contain the author's name and journal name, e.g. `authorname_procb_ESM_figures.pdf`

All supplementary materials accompanying an accepted article will be treated as in their final form. They will be published alongside the paper on the journal website and posted on the online figshare repository. Files on figshare will be made available approximately one week before the accompanying article so that the supplementary material can be attributed a unique DOI. Please see: <https://royalsociety.org/journals/authors/author-guidelines/>

4) Data-Sharing and data citation

It is a condition of publication that data supporting your paper are made available. Data should be made available either in the electronic supplementary material or through an appropriate repository. Details of how to access data should be included in your paper. Please see <https://royalsociety.org/journals/ethics-policies/data-sharing-mining/> for more details.

<http://datadryad.org/submit?journalID=RSPB&manu=RSPB-2021-2358> which will take you to your unique entry in the Dryad repository.

Once again, thank you for submitting your manuscript to Proceedings B and I look forward to receiving your final version. If you have any questions at all, please do not hesitate to get in touch.

Sincerely,
Professor Hans Heesterbeek
mailto:proceedingsb@royalsociety.org

Associate Editor

Comments to Author:

The authors have done an excellent job of addressing each of the reviewers' concerns. The manuscript is significantly revised and considerably improved. The research is important and will impact our understanding of pathogen interactions.

Reviewer(s)' Comments to Author:

Referee: 1

Comments to the Author(s).

The authors have adequately addressed the concerns of both reviewers. The changes made by the authors have improved the clarity of the writing and figures; the realism of their simulations by adding pre-existing immunity to influenza; and the impact of their results through the additional seasonal sensitivity analysis. The manuscript and context are important for both the scientific community and for public health, as their results imply that studies using test-negative designs to infer directionality and magnitude of viral interactions may be flawed under certain conditions.

Referee: 2

Comments to the Author(s).

Sorry for the delay. I agree with the author's answers and I am therefore happy to recommend the publication of this manuscript.

Decision letter (RSPB-2021-2358.R1)

06-Dec-2021

Dear Dr Domenech de Cellès

I am pleased to inform you that your manuscript entitled "The pitfalls of inferring virus-virus interactions from co-detection prevalence data: Application to influenza and SARS-CoV-2" has been accepted for publication in Proceedings B.

Data Accessibility section

Open Access

Paper charges

Sincerely,

Appendix A

Associate Editor

Board Member: 1

Comments to Author:

Following two expert reviews, there is consensus that the manuscript submitted by Domenech de Cellès et al. provides important insights to the limitations of current methods to measuring virus-virus interactions. This is undoubtedly an important topic. All reviewers agree that the manuscript would benefit from major revision. In preparing a revised manuscript for resubmission, please address carefully each point raised by the reviewers.

We thank the editor and our two reviewers for their comments and their thorough reading of our manuscript. In response, we have substantially revised the manuscript and conducted new analyses to improve the strength of our results and to extend the scope of our model. Specifically:

1. We now make the more realistic assumption of pre-existing immunity to influenza infection, captured by the initial fraction recovered and fixed to $r_0 = 0.4$ (1). In addition, in figure 3 we now assume a symmetric (on a log-scale) interval 0.2–5 for the interaction parameter θ and log-transform both axes. We re-ran the entire influenza–SARS-CoV-2 analysis and updated the corresponding figures 2–4 accordingly (please note that the former figure 2 was moved to the supplement is now figure S1, figures 3–4 are now figures 2–3). We believe these changes not only add biological realism, but also increase the scope of our model to emerging *and* seasonal viruses.
2. We substantially revisited the global sensitivity analysis. First, we added the new parameter r_0 to the list of inputs. Second, we calculated and plotted the parametric uncertainty of the GAMs (updated figure 4). Finally, to better understand the association of the prevalence ratio with each input parameter, we conducted a new analysis to estimate the effect of each parameter on the numerator (prevalence of co-infection p_{12}) and the denominator (product of single-infection prevalences $p_1 p_2$) of the prevalence ratio. This new analysis is now described in the Methods (in subsection “Global sensitivity analyses”), and its results presented in the new figure S5 and in the Results (subsection “Global sensitivity analysis highlights properties of viral infection that obscure or facilitate estimation of interaction”).
3. We conducted a new sensitivity analysis to assess how seasonal transmission affected the ability of the prevalence ratio to infer interaction. This new analysis is described in the Methods (subsection “Sensitivity analyses for influenza and SARS-CoV-2”) and its results presented in the new figure S4 and in the Results (subsection “Sensitivity analyses demonstrate the results’ robustness for influenza and SARS-CoV-2”). Assuming that the peak time of seasonal forcing approximately coincides with the peak time of co-infection, we find that interaction is more severely

underestimated by the prevalence ratio for higher amplitude of seasonal forcing. This result further reinforces our main conclusions regarding the shortcomings of the prevalence ratio.

A detailed, point-by-point response to the other reviewers' comments is provided below.

Reviewer(s)' Comments to Author:

Referee: 1

Comments to the Author(s)

The reviewed article focused on the ability for co-detection prevalence data to inform virus-virus interactions, using a pair of viruses that cause seasonal epidemics (influenza) and a current pandemic (SARS-CoV-2). Research has shown that respiratory viruses interact, leading to both positive and negative interactions, which can alter dynamics of co-infections. Early epidemiological studies investigating viral interactions between SARS-CoV-2 and a variety of respiratory viruses used the prevalence ratio to estimate the direction, and more importantly, strength of interactions among viruses. However, there is limited support for prevalence ratios to accurately estimate interactions among viruses (and all parasites). This study utilized simulations of a SEITR disease model with varying transmission parameters to test whether the test-negative design (co-detection prevalence data) is valid for testing for positive/negative interactions between respiratory viruses. The authors state that their results show that caution should be taken when interpreting epidemiological studies of SARS-CoV-2 (and other viruses) that use the test-negative design to infer directionality and magnitude of viral interactions.

The authors provide a clear background of viral interactions as well as current/past approaches to investigate interactions between SARS-CoV-2 and other respiratory viruses. The results of this study are not surprising, given the work of Fenton et al. 2014, but are exceedingly important to be reiterated, as early SARS-CoV-2 manuscripts make broad claims that may not be supported using the prevalence ratio, as shown here. I have one major concern regarding the interpretation of the relationship between the prevalence ratio and the authors' measure of interaction strength (θ), but find the rest of the manuscript and study to be satisfactory. The manuscript is well organized and well written, and the figures are clear and easy to interpret.

Comments

Overall: The authors discuss the limitations of the prevalence ratio in terms of its ability to estimate strength of interactions. Specifically, the authors state that “because of a concave association, under-estimation became more severe as the strength of the interaction increased”. While this concave relationship is apparent and causes the inherent underestimation of interaction strength, the concavity of this relationship is caused not by the prevalence ratio, but rather your measure of interaction strength (θ , theta), because θ (theta) itself is not symmetrical. Specifically, negative interactions are inherently bound between 0 and 1, while positive interactions are all numbers above 1. Therefore, a natural log transformation of θ results in a symmetrical distribution of interactions and would be akin to a log response ratio (sensu a meta-analysis). A natural log transformation of a 2x positive interaction (θ (theta) = 2) is 0.693, while a natural log transformation of a 2x negative interaction (θ (theta) = 0.5) is -0.693; and natural log transformation of unity (1) is 0. A 1:1

slope (and intercept of 1) would then indicate whether the prevalence ratio accurately measures strength of interactions among viruses.

I understand the purpose of this paper is to highlight the shortcomings of the PR in inferring interactions, *sensu* Fenton et al 2014, and I generally agree with the authors' conclusions that PR is not as informative as it seems and, instead, longitudinal studies are needed. However, the authors need to address the ability of the PR to accurately predict sign and strength of interactions after remediating the non-symmetry of their measure of interaction strength (θ , theta).

We thank the reviewer for this comment and for their answer to our query while we were preparing the revisions. In response (see also our response to the editor above), we now consider a log-symmetric interval (0.2–5) for the interaction parameter θ and we log-transform the x -axis in figure 2. Please note that we also decided to log-transform the y -axis, so that 1) the identity 1:1 line did not appear as a curve; and 2) the distance between the identity line and the curves quantified the relative under-estimation of interaction by the prevalence ratio. While discussing this figure, we no longer mention a concave association but simply indicate that the degree of under-estimation increases with the strength of interaction and with the duration of the post-infectious period (Methods, subsection “The prevalence ratio correctly identifies the sign, but not the degree, of uniform interactions”).

Line 22-23: Is there evidence that SARS-CoV-2 causes seasonal epidemics? Currently, still in pandemic, with potential for seasonal epidemic, but not sure there is enough evidence to argue this at this point.

This is an excellent point, also raised by reviewer 2. In response, we modified this sentence (Abstract): “To examine the validity of this assumption, we designed a simulation study that built on a broadly applicable epidemiological model of co-circulation of two emerging or seasonal respiratory viruses.” Please note that the revised model now allows some degree of pre-existing immunity (see our response to the editor above), so that our model now applies to both emerging and seasonal respiratory viruses.

Lines 49-51: Please clarifying the directionality of the suggested indirect effects of vaccines on non-target pathogens.

To clarify, we revised this sentence as follows (Introduction, end of first paragraph): “Indeed, vaccines that directly target a pathogen may also indirectly affect non-target pathogens that interact with this target pathogen—an effect expected (by the law of signs) to reduce the non-target pathogen burden in case of positive interactions, and to increase it in case of negative interactions.”

Lines 142-146: You should also test a second alternative hypothesis of non-symmetric, allow the direction of the effect of SARS-CoV-2 on influenza to be the opposite sign of the effect of influenza on SARS-CoV-2.

The reviewer raises an interesting point. We initially decided to restrict our analysis to interactions of the same sign (see Methods/Model parametrization), for two reasons. First, to the best of our knowledge we are not aware of a 2-pathogen system with opposite signs of interaction—although, as the reviewer correctly points out, this could theoretically be the case for influenza and SARS-CoV-2. More importantly, second, assuming opposite signs of interaction would make the classification of the overall interaction—and the subsequent comparison with the prevalence ratio—difficult. As a result, we tested only the “limit” case of neutral–negative (or neutral–positive) interactions, classified as globally positive (or globally negative). Experimental evidence in animal models suggests that such systems exist, for example influenza and RSV (i.e., influenza interferes with RSV, but RSV does not interfere with influenza) (2). That said, although beyond the scope of this study, opposite-sign interactions may represent an interesting topic of further research. Possibly, such interactions may be difficult to detect (even with longitudinal data), as they could “cancel out” the signal in epidemiological data.

Line 137: Change 5×10^{-2} to 0.05 days.

Done.

Line 171-172: The authors need to provide mean \pm se for the “peak time” of co-infection across scenarios.

We revised this sentence as follows (Results/The prevalence ratio correctly identifies the sign, but not the degree, of uniform interactions): “In all scenarios, however, the peak time was approximately identical (range: 77.7–79.7 days).” Please note that, because we now assume pre-existing immunity to influenza, the simulations displayed in figure S1 and the numerical values of peak co-infection prevalence have slightly changed.

Line 203: Only the positive interaction is “severely under-estimated” in this scenario, the negative interaction may be over-estimated, or possibly accurately estimated. But see general comment above regarding assessing the PR ability to estimate interaction strength.

We now write (Results/Sensitivity analyses demonstrate the results' robustness for influenza and SARS-CoV-2): “However, the strength of positive interaction was also more severely under-estimated in this scenario.” Please note that, because we now assume pre-existing immunity to influenza, figures 3, S2, and S3 have slightly changed compared with the initial submission.

Line 212-216: The authors posit that parameters in their SEITR model may differentially affect single- and co-infections and state that parameters associated with interactions post-infection affect single-infection prevalences more rapidly and strongly than co-infection, but provide no evidence for these statements. The authors need to present the relationships among these post-infection parameters and single-infection prevalences in conjunction with those of co-infection prevalence to then compare effects of these parameters quantitatively on

the single- and co-infection endpoints. Or, at least, move these lines to the discussion, as they seem more like assumptions than supported by evidence within this manuscript.

We agree with this comment, and we have removed this text from the manuscript. Instead, in the revised submission we conducted a new analysis to dissect the effect of each input parameter on the numerator (prevalence of co-infection) and the denominator (product of single-infection prevalences) of the prevalence ratio. The results of this new analysis are shown in the new figure S5 and discussed in the Results (subsection “Global sensitivity analysis highlights properties of viral infection that obscure or facilitate estimation of interaction”). We find that both the numerator and the denominator vary monotonically with each parameter, but at a different rate—thereby explaining the overall effect on the prevalence ratio. Although not a full, “mechanistic” explanation, we believe this new analysis sheds some light on these associations and on the parameters that sensitively shape the dynamics of viral co-circulation.

Line 237: define “low” prevalence.

We revised this sentence as follows, based on the peak co-infection prevalence for positive interactions in figure 2 (Discussion, second paragraph): “Besides the prevalence ratio, we found that the prevalence of co-infection was also an unreliable measure of interaction, as low prevalences ($\leq 2.8\%$ in Fig. 2, bottom panel) could be consistent with strong, positive interactions.”

Line 238-240: What deficiencies do you refer to? Low prevalence ratios? Further, the sensitivity analysis indicates that for a virus with a higher reproduction number (e.g., SARS-CoV-2) there is no relationship between prevalence ratio and latent period, suggesting that latent period may not affect co-infection. Does this suggest that the length of latent period has no affect on co-infection prevalence?

This is exactly right. In the global sensitivity analysis, we indeed found that higher values of the reproduction number not only decreased the prevalence ratio, but also dampened its association with the other input parameters (figure 4). Hence, this property (high reproduction number) alone may decrease the ability of the prevalence ratio to infer SARS-CoV-2 interactions. We have clarified this sentence as follows (Discussion, second paragraph): “As suggested by our global sensitivity analysis, the deficiencies of statistics based on co-infection prevalence may be even more severe for SARS-CoV-2 infection, characterized by a relatively high reproduction number.”

Line 259: Your results indicate that PR does more often than not predict the direction of interaction. Please provide estimates of the percent of runs in which PR accurately predicted the direction of interactions (e.g., PR correctly identified the direction of 70% of positive interactions, but only 50% of negative interactions).

We added this information (11% of scenarios with incorrect sign inferred for negative interactions and 14% for positive interactions), as required (Results, subsection “Higher

interaction post-infection can cause the prevalence ratio to misidentify non-uniform interactions”).

Figure 5: For ease of interpretation, provide symbols with titles on each panel. The text consistently refers to these parameters by their symbol, rather than what each parameter corresponds with. Even with Table 1, it takes effort to match symbols with parameter definitions. Additionally, these lines represent the relationship between PR and each input parameter, as estimated via a GAM. As such, these figures either need data points or error bars from the GAMs to provide a measure of model accuracy and clarity that these lines are from statistical models, not the SEITR model.

In response to this comment, we substantially revised this figure. The symbol of each parameter now appears in the corresponding panel title, and the 99% confidence intervals from the GAMs are plotted as ribbons surrounding the estimates. To make clear that the curves come from statistical models, we also changed the title of the y-axis (to “Estimated prevalence ratio”) and updated the legend of the figure.

Referee: 2

Comments to the Author(s)

This paper is focusing on the possibility to use co-detection ratio as a proxy for inferring pathogen/pathogen interactions, with a specific focus on Sars-Cov-2 and influenza. Briefly, the authors shown that this method is not really reliable. I think this is an interesting paper that would deserve to be published once my comments would have been addressed.

My main concern is about the context simulated, which is focusing actually on two emerging pathogens (since only transient dynamics is considered). However, even if influenza is more complicated than that, we cannot really assume that everybody is susceptible to influenza. More broadly, it rises the question about the validity of these results when considering endemic pathogens instead of emerging pathogens. This is very important to clarify regarding the strength of the messages ("In conclusion, our results show that the inherently complex, non-linear dynamic of respiratory viruses makes the interpretation of seemingly intuitive measures of interaction difficult, if not impossible.") and the pathogens studied as a case-study (sars-cov-2 and influenza).

We thank our reviewer for this valuable comment. Following a previously described model of influenza transmission (3), we initially assumed that pre-existing immunity to influenza could be captured by the initial reproduction number. However, we now realize this assumption was not necessary, and we modified the model to allow for pre-existing immunity (fixed to 40% for influenza [(1)] and 0% for SARS-CoV-2). We re-simulated all the models and updated figures 2–3 accordingly. In addition, we re-ran the global sensitivity analysis to incorporate this new parameter, which turned out to sensitively affect the prevalence ratio (see updated figure 4 and our response to the editor above). As pointed out by the reviewer, we believe these changes substantially increase the scope of our model, which can now be applied not only to emerging viruses (like SARS-CoV-2), but also to seasonal viruses (like influenza, RSV, rhinoviruses, etc.).

My other concern is about the model itself, especially the lack of transmission seasonality (not required for emerging pathogens, but could be needed for endemic ones) and the fact that stochasticity is not considered (but since the co-detection method does not seem to work very well, adding stochasticity will probably makes this result even stronger). I think these two important features need to be, at least, discussed.

The reviewer's point on stochasticity is exactly right. Indeed, our deterministic model can be interpreted as a best-case scenario, from which any deviation (like stochasticity caused by error measure) is expected to worsen the ability of the prevalence ratio to infer interaction. We discussed this important limitation in the original submission (Discussion, fifth paragraph): "First, because we used a deterministic model expressed in proportions, we sidestepped the important issue of statistical uncertainty, caused for example by finite sample size or imperfect measurement of infection prevalences. As the prevalence ratio was found to

systematically under-estimate the strength of interaction, such uncertainty—inevitable in practice—may further limit the ability of the prevalence ratio to correctly identify interactions.”

The other reviewer’s comment on transmission seasonality is also exactly right. In practice, as most respiratory viruses circulate during winter in temperate climates, we expect that seasonal drivers like weather will simultaneously affect their dynamics and therefore the prevalence ratio. To test this scenario, and in response to the reviewer’s comment, we conducted a new sensitivity analysis. Specifically, we added seasonal transmission forcing, modeled as a sine wave with different peak amplitudes and peak time coinciding with the peak time of co-infection (as noted above, the most likely situation during winter in temperate climates). This new analysis is detailed in the Results (subsection “Sensitivity analyses for influenza and SARS-CoV-2”) and its results presented in the new figure S4 and in the results (subsection “Sensitivity analyses demonstrate the results' robustness for influenza and SARS-CoV-2”). We found that the prevalence ratio was moderately sensitive to the amplitude of seasonality, with interaction more severely under-estimated for higher amplitude. This result further reinforces our main conclusions regarding the shortcomings of the prevalence ratio.

References

1. Kramer SC, Shaman J. Development and validation of influenza forecasting for 64 temperate and tropical countries. *PLoS Comput Biol.* 2019 Feb;15(2):e1006742.
2. Chan KF, Carolan LA, Korenkov D, Druce J, McCaw J, Reading PC, et al. Investigating Viral Interference Between Influenza A Virus and Human Respiratory Syncytial Virus in a Ferret Model of Infection. *J Infect Dis.* 2018 Jul 2;218(3):406–17.
3. Chowell G, Miller MA, Viboud C. Seasonal influenza in the United States, France, and Australia: transmission and prospects for control. *Epidemiol Infect.* 2008 Jun;136(6):852–64.